# High Dual Expression of the Biomarkers CD44v6/α2β1 and CD44v6/PD-L1 Indicate Early Recurrence after Colorectal Hepatic Metastasectomy

**DOI:** 10.3390/cancers14081939

**Published:** 2022-04-12

**Authors:** Friederike Wrana, Katharina Dötzer, Martin Prüfer, Jens Werner, Barbara Mayer

**Affiliations:** 1Department of General, Visceral, and Transplant Surgery, Ludwig-Maximilians-University, Marchioninistraße 15, 81377 Munich, Germany; friederike.schlueter@med.uni.muenchen.de (F.W.); katharina.doetzer@muenchen-klinik.de (K.D.); martin.pruefer@dritter-orden.de (M.P.); jens.werner@med.uni-muenchen.de (J.W.); 2German Cancer Consortium (DKTK), Partner Site Munich, Pettenkoferstraße 8a, 80336 Munich, Germany

**Keywords:** colorectal cancer, liver metastases, lung metastases, protein biomarker, dual expression, early recurrence, poor prognosis

## Abstract

**Simple Summary:**

Distant metastasis in colorectal cancer still correlates with poor prognosis, emphasizing the high need for new diagnostic and therapeutic strategies. In the present study, liver and lung metastases revealed profound differences in the expression pattern of metastasis-driving protein biomarkers. This suggests the adaption of the therapy to the biology of the metastatic organ site. High expression of the cell adhesion molecule CD44v6 and high dual expression of CD44v6, combined with the cell adhesion molecules integrin α2β1, as well as the checkpoint inhibitor molecule PD-L1, correlated significantly with early recurrence after hepatectomy, in a substantial number of liver metastatic patients. These findings suggest the need for the implementation of biological risk factors into clinical risk scores, aiming to make the prognosis of the individual patient more precise. Further, dual expression of protein biomarkers that are druggable, such as CD44v6/α2β1 and CD44v6/PD-L1, can identify high-risk patients for targeted therapy that might provide a survival benefit.

**Abstract:**

Considering the biology of CRC, distant metastases might support the identification of high-risk patients for early recurrence and targeted therapy. Expression of a panel of druggable, metastasis-related biomarkers was immunohistochemically analyzed in 53 liver (LM) and 15 lung metastases (LuM) and correlated with survival. Differential expression between LM and LuM was observed for the growth factor receptors IGF1R (LuM 92.3% vs. LM 75.8%, *p* = 0.013), EGFR (LuM 68% vs. LM 41.5%, *p* = 0.004), the cell adhesion molecules CD44v6 (LuM 55.7% vs. LM 34.9%, *p* = 0.019) and α2β1 (LuM 88.3% vs. LM 58.5%, *p* = 0.001) and the check point molecule PD-L1 (LuM 6.1% vs. LM 3.3%, *p* = 0.005). Contrary, expression of HGFR, Hsp90, Muc1, Her2/neu, ERα and PR was comparable in LuM and LM. In the LM cohort (*n* = 52), a high CD44v6 expression was identified as an independent factor of poor prognosis (PFS: HR 2.37, 95% CI 1.18–4.78, *p* = 0.016). High co-expression of CD44v6/α2β1 (HR 4.14, 95% CI 1.65–10.38, *p* = 0.002) and CD44v6/PD-L1 (HR 2.88, 95% CI 1.21–6.85, *p* = 0.017) indicated early recurrence after hepatectomy, in a substantial number of patients (CD44v6/α2β1: 11 (21.15%) patients; CD44v6/PD-L1: 12 (23.1%) patients). Dual expression of druggable protein biomarkers may refine prognostic prediction and stratify high-risk patients for new therapeutic concepts, depending on the metastatic location.

## 1. Introduction

According to international guidelines [1,2,3], metastasectomy currently offers the best chance for long-term survival for selected colorectal cancer patients. Additional standard chemotherapy for patients with resectable liver metastases resulted in the prolongation of disease-free survival (DFS) and progression-free survival (PFS) but revealed no significant improvement in overall survival (OS) [4,5]. In patients with resectable pulmonary metastases, the outcome of peri-operative chemotherapy is inconclusive [6,7]. However, despite curative-intent metastasectomy, more than half of the patients suffer recurrence [8,9]. This highlights the urgent need for the implementation of new strategies to identify high-risk patients suitable for personalized therapy, aiming to improve treatment outcome and survival [10].

Colorectal cancer preferentially metastasizes to the liver, followed by the lung and the peritoneum and, more rarely, in bone, ovary and the brain [11,12,13]. The metastatic pattern depends on the sidedness of the primary colorectal tumor. Elucidating the underlying mechanisms of the metastatic organotropism, profound molecular differences were observed between right-sided and left-sided CRC cancers. Similarly, the tumor microenvironment seems to have a deep impact on the metastatic site [14]. Indeed, for primary metastatic colorectal cancer, a growing body of molecular data is available, resulting in the continuous development of targeted therapies and improvement in survival [15,16].

Comparative analysis of primary CRC and corresponding metastatic sites revealed maintenance of the main driver mutations in both liver and lung metastases, some of which are approved for CRC therapy, such as RAS, BRAF and MSI [17,18,19]. In contrast, genomic [20,21,22], transcriptomic [23] and proteomic [24] profiling identified molecular differences between primary tumor, liver and lung metastases that might have potential therapeutic implications for specific metastatic sites. Moreover, distant metastases in different organs revealed discordant responses to standard chemotherapy [25], all together, supporting the concept of inter- and intratumor heterogeneity, which is one of the key factors in tumor progression, therapeutic resistance, and poor patient outcome.

In the present study, a panel of protein biomarkers was selected, which drive the complex metastatic process of primary colorectal cancer and lead to poor prognosis. In contrast, little information is available on the expression pattern of these prognostic factors in liver and lung metastases. The protein biomarker panel encompassed the growth factor receptors epidermal growth factor receptor (EGF-R) and hepatocyte growth factor receptor (HGF-R) [26], human epidermal growth factor receptor (Her2/neu) [27], insulin-like growth factor 1 receptor (IGF-1R) [28], estrogen receptor alpha (Erα) [29] and progesterone receptor (PR) [30], the cell adhesion molecules CD44v6 [31], Muc1 [32] and integrin α2β1 [33], the chaperone heat shock protein 90 (Hsp90) [34], and the immune checkpoint molecule programmed death ligand 1 (PD-L1) [35]. Interestingly, the protein biomarkers selected are drug targets, for which drugs are already approved or for which clinical trials are ongoing, in primary colorectal cancer or other cancer types. This could open up new options for second and further line treatments in colorectal cancer.

The present study aimed (1) to identify the phenotypic heterogeneity in tumor biology between colorectal liver and lung metastases and (2) to stratify patients with a high risk for early recurrence after hepatic metastasectomy.

## 2. Materials and Methods

### 2.1. Patient Cohort

The patient cohort consists of 68 patients with metastatic colorectal cancer, receiving metastasectomy with curative intent at the Department of General, Visceral, and Transplant Surgery, Ludwig-Maximilians-University, Munich, Germany. A liver metastasis (LM, *n* = 53) or a lung metastasis (LuM, *n* = 15) was analyzed from each patient. Double-coded tissues and the corresponding data used in this study were provided by the Biobank of the Department of General, Visceral, and Transplant Surgery, Ludwig-Maximilians-University Munich, Munich, Germany. This Biobank operates under the administration of the Human Tissue and Cell Research (HTCR) Foundation. The framework of HTCR Foundation, which includes obtaining written informed consent from all donors, has been approved by the ethics commission of the Faculty of Medicine at the LMU (approval number 025-12) as well as the Bavarian State Medical Association (approval number 11142) in Germany. All liver metastases were diagnosed as the first relapse of the individual patient. Lung metastases represented first (*n* = 3), second (*n* = 8) and later stage relapse (*n* = 4). Survival analysis was performed for 52 patients diagnosed with liver metastases. One patient was lost to follow up. Follow-up period of the patient cohort was from December 2010 until February 2018.

### 2.2. Immunohistochemistry and Evaluation of Biomarker Expression

Fresh tumor samples including adjacent benign reference tissue were collected according to international biobanking standards. After surgery the tumor samples were immediately snap frozen in liquid nitrogen. Serial cryosections (5 µm) were performed and air dried over night at room temperature. Sections were either fixed in acetone, or for the ERα und PR staining in formalin solution (10%). Immunohistochemistry was performed using the standard avidin-biotin-peroxidase complex method [36,37,38]. Briefly, unspecific Fc receptors were blocked with 10% AB-serum in D-PBS, pH 7.4 for 20 min. Endogenous biotin was blocked using the Avidin-/Biotin-blocking Kit for 15 min. The primary antibodies (Table 1) were incubated for one hour. Some antibodies were detected with the secondary biotinylated antibody (111-065-114; wc 7.0 µg/mL; JacksonImmunoResearch, West Grove, PA, USA for anti-rabbit and 315-065-048; wc 0.75 µg/mL; JacksonImmunoResearch for anti-mouse) for 30 min, followed by the peroxidase-conjugated streptavidin (016-030-084; wc 1.0 µg/mL; affymetrix eBiosciences, Santa Clara, CA, USA) for another 30 min. Other primary antibodies were detected with the amplification Kit ZytoChem Plus (HRP060; Zytomed Systems, Bargteheide, Germany) according to the instructions of the manufacturer (marked in Table 1 with Kit: +). For visualization of the antigen–antibody reaction all slides were developed in a 3-Amino-9-ethylcarbazole solution containing 35% hydrogen peroxide (AEC staining) for eight minutes in darkness. Counterstaining was performed with Mayer’s hemalum solution. All incubation steps were performed in a humid chamber at room temperature. Specificity of the staining was controlled by the corresponding isotype controls (Table 1). Cancer cells were visualized by EpCAM and pan-cytokeratin expression.

For the evaluation of biomarker expression, the size of the measurement field was standardized using a normalized grid at 100× magnification (Olympus microscope BX50, Olympus, Hamburg, Germany). The biomarker-positive tumor area was determined in relation to the total tumor area. The percentage of biomarker-positive tumor cells was expressed by semiquantitative estimation in 10% increments. Staining results were evaluated by two independent observers (FW, BM). External monitoring was performed by local pathologists (Institute of Pathology, LMU Munich, Munich, Germany, T. Kirchner) and for Her2/neu expression by J. Rüschoff (Institute of Pathology Nordhessen, Kassel, Germany, Rüschoff) [39].

For some biomarkers standardized cut-off values are given, namely ERα and PR [40], Her2/neu [39,41], Muc1 [42,43], and PD-L1 [36,44]. In the absence of standardized cut-offs for other biomarkers, cut-offs were assessed using the biphasic distribution, which was statistically defined using the mean antigen expression in liver or lung metastases. Biomarker expression below the calculated cut-off was defined as low expression, and biomarker expression above the calculated cut-off was defined as high expression. The same cut-off values were used for single biomarker analysis and the evaluation of dual biomarker expression. In addition to the tumor tissue, antigen expression was evaluated on the adjacent benign liver and lung tissues.

### 2.3. Statistical Analysis

All statistical analyses were performed with IBM SPSS v. 23. Mean biomarker ex-pression between liver and lung metastases was compared using the Mann–Whitney U-test. The prognostic impact of single and dual biomarker expression was evaluated using Kaplan–Meier analysis (log rank test, ‘pairwise over strata’) and multivariate Cox regression analysis (biomarker expression used as ‘categorical covariate’, ‘First’ as reference category). OS was defined as the time from metastasectomy until the last follow-up or death of the patient. PFS was defined as the time from metastasectomy until the next progression. A *p*-value of ≤0.05 was considered as significant.

## 3. Results

### 3.1. Patient Characteristics

In the present study, 53 liver metastases and 15 lung metastases surgically resected from colorectal cancer patients were analyzed. Men were more frequently affected than women (LM: ratio 1.79:1; LuM: ratio 4:1). Most (66.04%) liver metastases were detected at primary diagnosis (synchronous), whereas all lung metastases were documented at a later time (metachronous). Liver and lung metastases were diagnosed as single organ metastases. However, at the organ site, tumor disease was frequently extensive (number of nodules within the metastatic organ >1; LM: 64.15%, LuM: 53.33%; multilobular involvement; LM: 56.6%, LuM: 66.67%). Still, most patients were resected with curative intent (R0; LM: 73.58%, LuM: 80%). Further, 32 of 53 (60.38%) patients diagnosed with liver metastases received first-line chemotherapy (5-FU as single agent: 34.38%, oxaliplatin-based: 43.75%, irinotecan-based: 15.63%, others: 6.25%) and 23 of 53 (43.40%) received neoadjuvant chemotherapy before liver metastasectomy. Of these, 10 of 15 (66.67%) patients were treated with front line chemotherapy (5-FU as single agent: 10%, oxaliplatin-based: 80%, others: 10%) and 8 of 15 (53.33%) patients received neoadjuvant chemotherapy, right before surgery of the lung metastasis studied. Complete treatment records were not available for all patients with lung metastases.

Patient characteristics are summarized in detail in Table 2.

Survival analysis was performed in the patient cohort with liver metastases but was omitted in patients with lung metastases because of small sample size. Patients diagnosed with multiple (>1) LM had a significantly shorter PFS compared to patients diagnosed with a single liver metastasis (multiple metastases, PFS: 6.5 months; single metastasis, PFS: 10 months; log-rank, *p* = 0.014). Patients with synchronous LM relapsed much faster compared to patients with metachronous LM (synchronous, PFS: 7 months; metachronous, PFS: 16 months; log rank, *p* = 0.001). None of the patient characteristics revealed an impact on OS.

### 3.2. Differential Biomarker Expression in Colorectal Liver and Lung Metastases

Liver and lung metastases were comparatively analyzed with a panel of metastasis-related protein biomarkers. A differential expression pattern between liver and lung metastases was observed for the growth factor receptors IGF-1R (LuM 92.3% vs. LM 75.8%, *p* = 0.013) and EGF-R (LuM 68% vs. LM 41.5%, *p* = 0.004), showing a significantly higher fraction of positive cancer cells in the lung metastases, respectively. Similar results were obtained for the cell adhesion molecules CD44v6 (LuM 55.7% vs. LM 34.9%, *p* = 0.019) and integrin α2β1 (LuM 88.3% vs. LM 58.5%, *p* = 0.001), as well as for the check point molecule PD-L1 (LuM 6.1% vs. LM 3.3%, *p* = 0.005). In contrast, no significant difference was observed for the growth factor receptor HGF-R and the chaperon molecule Hsp90, both showing a high fraction of positive cancer cells in almost all distant metastases. Conversely, all but one metastatic lesion were found negative for the hormone receptors ERα and PR. One individual liver metastasis demonstrated 30% ERα positive cancer cells. Moreover, in colorectal liver and lung metastases, a minor fraction of the cancer cells were found positive for the cell adhesion molecule Muc1 and growth factor receptor Her2/neu. In fact, only one liver metastasis (60% Her2/neu positive cancer cells) qualified for anti-Her2/neu therapy. The number of biomarker-positive lesions and the means of biomarker expression are given in Table 3. The distribution of biomarker expression is shown for liver and lung metastases (Figure 1).

Biomarker analysis showed most of the benign liver tissues positive for HGF-R, EGF-R, and Hsp90. IGF-1R and PD-L1 were detected in a fraction of benign liver samples (IGF-1R: 11 out of 52, 21.2%; PD-L1: 10 out of 52, 19.2%). Interestingly, benign liver tissue was negative for Muc1, CD44v6 and the integrin α2β1. In contrast, all biomarkers tested were detected on benign lung tissue, although the integrin α2β1 (10 out of 15, 66.6%) and Muc1 (8 out of 15, 53%) were observed on a reduced number of adjacent lung tissues. Data obtained in benign tissue samples are summarized in Appendix A. Figure 2 demonstrates the significantly different staining patterns by each biomarker of liver and lung metastases.

### 3.3. Prognostic Impact of Biomarker Expression in Colorectal Liver Metastases

The prognostic impact of the biomarkers was analyzed in patients with liver metastases. CD44v6, but none of the other biomarkers tested, was identified as an indicator for early recurrence. Liver metastases with a high fraction (>30%, *n* = 22) of CD44v6+ tumor cells significantly correlated with a shorter (median 7.0 months) PFS compared to LM with a low CD44v6 expression (≤30% CD44v6+ cells, *n* = 30; median 15.5 months; log rank *p* = 0.01). Recurrent liver metastases with a high proportion of CD44v6+ cancer cells showed more frequent multi-organ metastases (6 out of 19, 31.58%), compared to liver metastases with a low proportion of CD44v6+ cancer cells (3 out of 22, 13.65%). Almost all multi-organ metastases involved liver and lung, regardless of the extent of CD44v6 expression. Cox regression analysis confirmed the independent prognostic impact of CD44v6 on PFS (Table 4). No significant correlation was found between CD44v6 expression in LM and OS.

### 3.4. CD44v6-Related Dual Biomarker Expression in Colorectal Liver Metastases

Co-expression analysis was performed on CD44v6 and the metastasis-related biomarkers. Univariate analysis identified three pairs of highly expressed biomarkers associated with short PFS. Patients with liver metastases with strong expression of CD44v6 and integrin α2β1 showed a shorter mean PFS (3 months) compared to the group with only high expression of CD44v6 (7 months) (Table 5, Figure 3). Multivariate Cox regression analysis identified the combination of a high CD44v6 and a high integrin α2β1 expression (HR: 4.135, 95% CI: 1.648–10.375, *p* = 0.002) and the combination of a high CD44v6 and a high PD-L1 expression (HR: 2.882, 95% CI: 1.213–6.848, *p* = 0.017), as independent prognostic factors for short progression-free survival (Table 6). High co-expression was detected in a substantial number of patients; i.e., CD44v6 high (>30% positive tumors cells) combined with integrin α2β1 high (>80% positive tumor cells) in 11 out of 52 (21.15%) patients, CD44v6 high combined with Hsp90 high (>70% positive tumor cells) in 14 out of 52 (26.92%) patients and CD44v6 high combined with PD-L1 high (>1% positive cells) in 12 out of 52 (23.1%) patients.

## 4. Discussion

It is well published that primary colorectal cancer differs in its biology, depending on sidedness [45]. This also includes treatment-relevant characteristics, such as the RAS [46,47], MSI [48] and BRAF status [49]. In the present study, biomarker heterogeneity was identified between colorectal liver and lung metastases, namely for the cell adhesion molecules α2β1, CD44v6, the growth factor receptors IGF-1R, EGF-R and the immune checkpoint biomarker PD-L1. These site-specific differences in biomarker expression might reflect the complex multifactorial interactions between disseminated cancer cells and the target organ microenvironment [50]. Cancer cells with a unique tumor biology are homing to metastatic niches with a microenvironment promoting colonization, survival, and proliferation [51,52]. Liver and lung metastases reveal biological differences; for example, in the cellular composition of the microenvironment [36,52,53,54], the ECM signature [52,55,56] and the secretome profile [52,57]. Quantitative differences in protein biomarker expression were found between liver and lung metastases, showing a significantly higher proportion of IGF-1R-, EGR-R-, CD44v6-, α2β1-, and PD-L1-positive cancer cells in the lung. This observation confirms published data, showing a higher frequency of genetic drivers, such as KRAS alterations and MET amplification in lung metastases [20,58]. At the same time, lung metastases exhibit an increased immunosuppressive microenvironment and prometastatic inflammation [36,59]. These findings suggest distinct colonization mechanisms, involving both specific cancer cells with a higher propensity to metastasize to the lung and a lung-specific environment that facilitates metastasis of specific cancer cells. Targeting metastasis-relevant biomarker expression will open up new therapeutic opportunities, adjusted to specific metastatic localizations. This is in deep contrast to the current guideline, which recommends the concept of treating distant metastasis with the same therapy, independent from the metastatic organ site.

The protein biomarker expression pattern in liver metastases was tested for prognostic relevance. A high (>30%) fraction of CD44v6+ liver metastatic cells was identified as an independent prognostic factor mediating short progression-free survival. This finding supports CD44v6 as a metastatic driver. Multiple underlying molecular mechanisms have been described for CD44v6-mediated progression in colorectal cancer. Examples are interactions with the extracellular matrix components osteopontin and hyaluronic acid and the binding of different cytokines, such as HGF, EGF and VEGF [31,60]. Co-expression analysis identified two new independent risk factors associated with poor prognosis of CRC patients with liver metastases. Most interesting, high dual expression of CD44v6 and integrin α2β1 represents an indicator of early recurrence, defined as tumor relapse within six months after liver resection for colorectal metastases [61,62]. Direct and extracellular matrix-mediated molecular crosstalk between CD44v6 and various integrins, including α2β1, was found to promote cancer cell proliferation and invasion, tumor angiogenesis and chemoresistance, all involved in a considerable shortening of progression-free survival compared to the single CD44v6 expression [63,64,65]. In addition, dual expression of CD44v6 and PD-L1, indicating the crosstalk between tumor cells and the tumor microenvironment, significantly correlated with short survival. The subset of CD44v6+ colorectal cancers simultaneously expressing PD-L1 might represent stem-like properties and contributes to immune evasion mediating poor prognosis [66,67]. Similarly, co-mutations in RAS, TP53 and SMAD4, as well as in APC and PIK3CA, resulted in a worse outcome after hepatectomy compared to single mutations [19]. Therefore, our findings support the strategy of combining prognostic protein biomarkers to render the prediction of outcome more precise [68,69]. Further, these new factors might be included in clinical risk scores, similar as reported for the KRAS status in the GAME score [70] and the KRAS/NRAS/BRAF status in the CERR score [71], which resulted in the refinement to predict recurrence after resection of CRC liver metastases. In contrast to some of the most investigated therapeutic biomarkers, namely BRAF, MSI-high, and Her2/neu, all detected in a very small patient cohort [19,20], dual expression of the druggable targets CD44v6/α2β1 and CD44v6/PD-L1 was identified in about 20% of the liver metastatic patients.

In addition, these novel findings might have an impact on the development of new therapeutic strategies for liver metastatic CRC patients. Currently, new anti-CD44v6 treatment strategies, such as half antibodies conjugated nanoparticles [72], peptides (NCT03009214) and CD44v6-specific CAR gene-engineered T cells (NCT04427449, [73]) are under investigation and might also become a treatment option for CRC patients with CD44v6-positive liver metastases. Combination of two biomarkers might help to stratify patients more precisely for targeted therapy compared to single biomarker expression. For example, Shek et al., 2021, reported that only a subgroup of PD-L1-positive mCRCs responded to checkpoint inhibitor therapy [74]. In addition, dual expression of druggable biomarkers will further promote the promising concept of multiple target inhibition, aiming to improve treatment outcome and reduce the risk of drug resistance. Recently, the combination of the BRAF inhibitor Encorafenib with the EGF-R inhibitor Cetuximab has been reported as the new standard for the treatment of metastatic BRAF-mutated colorectal cancer [75]. Currently, a number of clinical trials are ongoing in advanced colorectal cancer, simultaneously inhibiting different targets. This includes combination therapy of the EGF-R inhibitor Panitumumab with the multi-kinase inhibitor Cabozantinib [76]. Further, anti-PD-L1 checkpoint inhibitors have been combined with targeted therapies, aiming to improve the response to immunotherapy [77]. In the present study, dual expression of PD-L1 and CD44v6 was found to correlate with poor prognosis and might represent a new therapeutic option for combination therapy. The second interesting pair of therapeutic targets identified in the present study was the co-expression of CD44v6 and the integrin α2β1. Both cell adhesion molecules were found to mediate chemoresistance [65,78]. Simultaneous inhibition of both targets might result in the circumvention of chemoresistance and represent a new anti-metastatic strategy of targeted therapy. Consideration of metastasis-driving protein biomarkers that predict early recurrence after hepatectomy might play a critical role in the clinical management of patients diagnosed with liver metastases [79]. The findings in the present study need to be confirmed in a larger, prospective trial.

## 5. Conclusions

A differential expression pattern of the druggable protein biomarkers α2β1, CD44v6, IGF-1R, EGF-R and PD-L1 was identified between colorectal liver and lung metastases. High expression of CD44v6, CD44v6/α2β1, and CD44v6/PD-L1 correlated significantly with early recurrence after hepatic metastasectomy. Dual biomarker expression may render the prognostic prediction more precise and stratify high-risk patients for new therapeutic concepts, depending on the metastatic organ site.

## Figures and Tables

**Figure 1 cancers-14-01939-f001:**
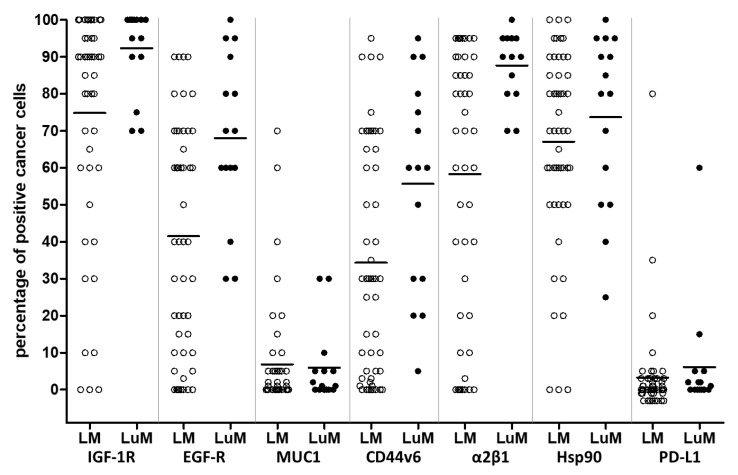
Biomarker Expression Pattern of Liver and Lung Metastases. horizontal bars, Means. Each dot represents a metastatic lesion; empty dots represent liver metastases (**LM**); filled dots represent lung metastases (**LuM**).

**Figure 2 cancers-14-01939-f002:**
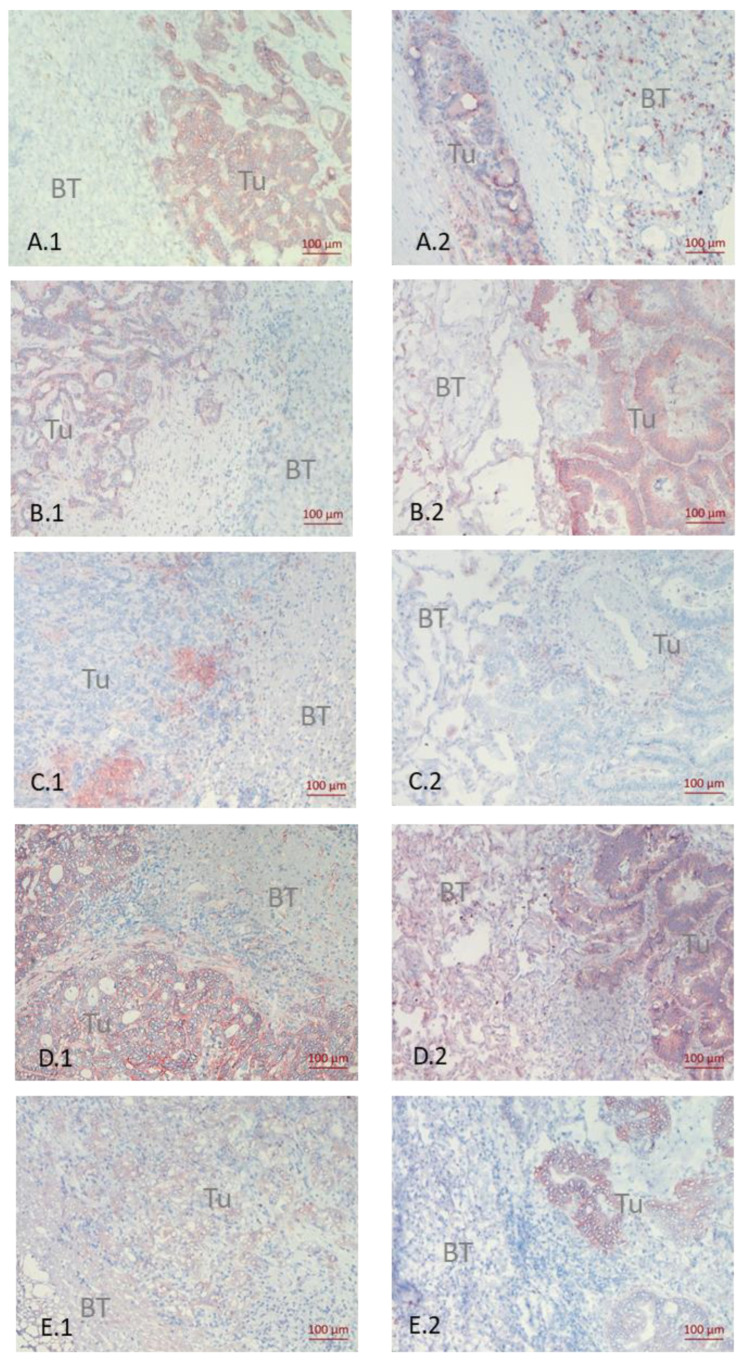
Immunohistochemical Staining of Different Biomarkers. Differential biomarker expression between liver (**1**) and lung (**2**) metastases demonstrated by immuno-histochemistry. (**A**), CD44v6; (**B**), α2β1; (**C**), PD-L1; (**D**), IGF-1R; (**E**), EGFR; **Tu**, tumor tissue; **BT**, Benign tissue.

**Figure 3 cancers-14-01939-f003:**
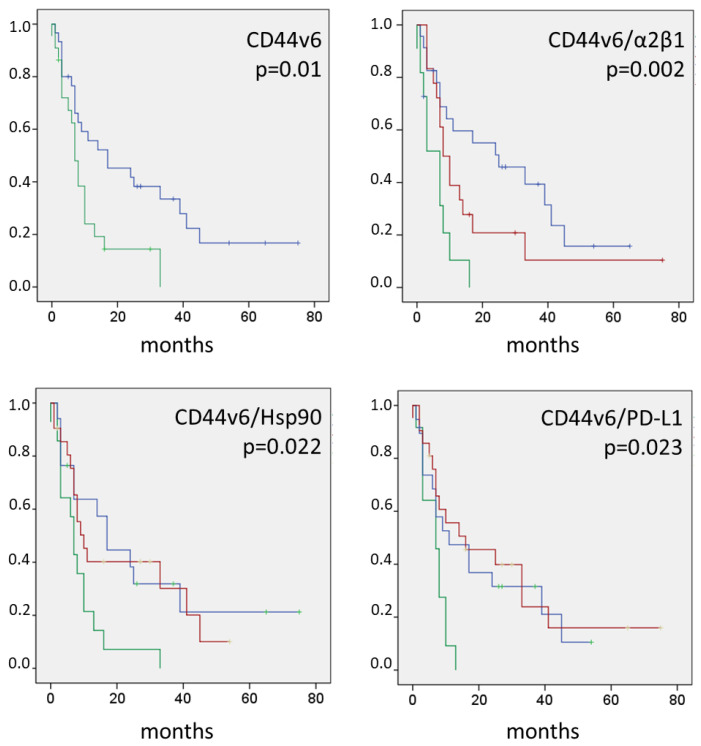
Kaplan–Meier Curves of CD44v6-Related Biomarker Expression in Colorectal Liver Metastases. **Blue lines**, low/low expression; **green lines**, high/high expression; **red lines**, high/low and low/high expression; log-rank *p*-values are given; cut-off values defining high- and low-level expression for the individual biomarker are given in Table 3.

**Table 1 cancers-14-01939-t001:** Antibody Panel for Immunophenotyping of Colorectal Liver and Lung Metastases.

Biomarker	Antibody/Clone	Species	Isotype	Working Concentration (µg/mL)	Kit	Source
HGF-R	Sp44	rabbit	IgG1	2.12	-	Spring Bioscience/Biomol, Pleasanton, CA, USA
IGF1-R	24–31	mouse	IgG1	4.0	+	Invitrogen, Carlsbad, CA, USA
EGR-R	H11	mouse	IgG1	2.94	-	Dako, Santa Clara, CA, USA
Her2/neu	4B5	rabbit	IgG1	1.5	-	Ventana, Roche, Basel, Switzerland
Erα	ID5	mouse	IgG1	2.5	+	Dako
PR	PgR 636	mouse	IgG1	2.5	+	Dako
Muc1	Ma55.2	mouse	IgG1	0.5	-	Monosan, Uden, The Netherlands
CD44v6	VFF-18	mouse	IgG1	1.0	-	eBioscience Affymetrix
α2β1	BHA2.1	mouse	IgG1	2.5	-	Millipore, Burlington, MA, USA
Hsp90	AC88	mouse	IgG1	10.0	+	Abcam, Cambridge, UK
PD-L1	MIH1	mouse	IgG1	10.0	+	Affymetrix
Positive controls						
Epcam	Ber-EP4	mouse	IgG1	5.0	-	Dako
Pan Cytokeratin	KL-1	mouse	IgG1	0.32	-	Zytomed Systems
isotype controls						
MOPC-21	MOPC-21	mouse	IgG1	5.0	-	Sigma-Aldrich, St. Louis, MO, USA
MOPC-21	MOPC-21	mouse	IgG1	4.0	+	Sigma-Aldrich
MOPC-21	MOPC-21	mouse	IgG1	10.0	+	Sigma-Aldrich
Rabbit mAb	DA1E	rabbit	IgG1	2.12	-	Cell Signaling, Danvers, MA, USA

**Table 2 cancers-14-01939-t002:** Patient Characteristics.

Parameters	Liver Metastases		Lung Metastases	
	*n*	%	*n*	%
**patient related**				
sex				
male	34	64.15	12	80.00
female	19	35.85	3	20.00
age (years)				
median	64		62	
mean	64		59	
range	30–89		37–74	
**metastasis related**				
grading				
G1/G2	39	81.25	11	73.33
G3	9	18.75	4	26.67
missing	5		0	
number of metastases *				
1	19	35.85	7	46.67
>1	34	64.15	8	53.33
diameter of the largest metastases (cm)				
median	3.5		1.8	
mean	4.29		2.25	
range	1.3–21.7		0.9–3.3	
type of metastasis				
synchronous	35	66.04	0	0.00
metachronous	18	33.6	15	100.00
R-status				
R0	39	73.58	12	80.00
R1	14	26.42	3	20.00
distinction of metastasis				
unilobular	23	43.4	5	33.33
multilobular	30	56.6	10	66.67
anatomical site				
left sided	7	13.21	7	46.67
right sided	15	28.30	8	53.33
both sided	31	58.49		
neoadjuvant chemotherapy ^#^				
yes	23	43.40	8	53.33
no	30	56.60	7	46.67
therapy options				
oxaliplatin-based	11	47.83	1	12.5
irinotecan-based	7	30.43	5	62.5
others	5	21.74	2	25.0

***n***, number of patients; **R-status**, residual status after surgery; *****, nodules within the metastatic organ; ^#^, administered directly before metastasectomy.

**Table 3 cancers-14-01939-t003:** Positivity and Distribution of Biomarkers in Liver and Lung Metastases.

Biomarker	Number of Positive Lesions				Number of Positive Cancer Cells (%)					Number of Positive Lesions above Cut-Offs				
	Liver		Lung		Median			Mean			Liver		Lung	
	*n* = 53	%	*n* = 15	%	Liver	Lung	*p*-Value	Liver	Lung	Cut Off *	*n* = 53	%	*n* = 15	%
HGF-R	52	98.1	15	100	95	95	0.166	87.7	95.3		n.t.			
IGF-1R	50	94.3	15	100	90	100	**0.013**	75.8	92.3	>80	29	54.7	12	80
EGF-R	45	84.9	15	100	40	70	**0.004**	41.5	68.0	>50	25	47.2	12	80
Her2/neu	19	35.8	8	53.3	0	1	0.575	5.7	1.7	>50	1	1.9	0	0
ERα	1	1.9	0	0	0	0	n.t.	0.6	0	≥1	n.t.			
PR	0	0	0	0	0	0	n.t.	0	0	≥1	n.t.			
Muc1	26	49.1	9	60	0	1	0.614	6.8	5.9	+/−	26	49.1	9	60
CD44v6	45	84.9	15	100	30	60	**0.019**	34.9	55.7	>30	23	43.4	10	66.7
α2β1	46	86.8	15	100	70	90	**0.001**	58.5	88.3	>80	20	37.7	11	73.3
Hsp90	51	96.2	15	100	75	80	0.475	68.7	73.9	>70	26	49.1	9	60
PD-L1	24	45.3	13	86.7	0	1	**0.005**	6.1	3.25	>1	24	45.3	11	73.3

***n***, number of patients; **n.t.**, not tested; *****, calculation of the cut-offs is given in the Materials and Methods Section.

**Table 4 cancers-14-01939-t004:** Multivariate Survival Analysis of CD44v6 Expression in Colorectal liver Metastases.

Variable	Groups	Cox Regression		
		HR	*p*-Value	95% CI
age (median in years)	>64/≤64	1.424	0.357	0.671–3.021
number of metastases *	>1/≤1	1.221	0.572	0.610–2.454
type of metastases	synchronous/metachronous	4.206	**0.004**	1.572–11.254
CD44v6 expression	>30%/≤30%	2.369	**0.016**	1.175–4.777

**HR**, Hazard ratio; *p*-value was calculated for progression free survival; **CI**, confidence interval; *****, nodules within the metastatic organ.

**Table 5 cancers-14-01939-t005:** Univariate Survival Analysis of CD44v6-Related Dual Biomarker Expression in Colorectal Liver Metastases.

Combination	Number of Patients (n)	Log Rank *p*-Value	Median PFS (month)
CD44v6 high *	22	**0.01**	7
CD44v6 low	30		15.5
CD44v6 high/IGF1-R high	15	0.142	7
CD44v6 high/IGF1-R low or CD44v6 low/IGF1-R high	20		9
CD44v6 low/IGF1-R low	17		17
CD44v6 high/EGF-R high	11	0.217	6
CD44v6 high/EGF-R low or CD44v6 low/EGF-R high	24		11.5
CD44v6 low/EGF-R low	17		9
CD44v6 high/Muc1 high	11	0.574	8
CD44v6 high/Muc1 low or CD44v6 low/Muc1 high	23		11
CD44v6 high/Muc1 low	18		7.5
CD44v6 high/α2β1 high	11	**0.002**	3
CD44v6 high/α2β1 low or CD44v6 low/α2β1 high	18		9
CD44v6 low/α2β1 low	23		24
CD44v6 high/Hsp90 high	14	**0.022**	7
CD44v6 high/Hsp90 low or CD44v6 low/Hsp90 high	21		9
CD44v6 low/Hsp90 low	17		17
CD44v6 high/PD-L1 high	12	**0.023**	7
CD44v6 high/PD-L1 low or CD44v6 low/PD-L1 high	21		14
CD44v6 low/PD-L1 low	19		11

**PFS**, progression-free survival; cut-off values defining high and low for the individual biomarker are given in Table 3; *****, calculation of the cut-offs is given in the Materials and Methods Section.

**Table 6 cancers-14-01939-t006:** Multivariate Survival Analysis of CD44v6-Related Dual Biomarker Expression in Colorectal Liver Metastases.

Variable	Groups	Cox Regression (PFS)		
		HR	*p*-Value	95% CI
age (median in years)	>64/≤64	1.561	0.256	0.724–3.366
number of metastases *	>1/≤1	1.398	0.358	0.684–2.855
type of metastases	synchronous/metachronous	3.813	**0.008**	1.407–10.332
CD44v6/α2β1 expression	high/high vs. low/low	4.135	**0.002**	1.648–10.375
	high/low and low/high vs. low/low	1.784	0.145	0.819–3.886
age (median in years)	>64/≤64	1.129	0.773	0.496–2.568
number of metastases	>1/≤1	1.321	0.460	0.632–2.762
type of metastases	synchronous/metachronous	3.345	**0.013**	1.289–8.680
CD44v6/Hsp90 expression	high/high vs. low/low	2.039	0.085	0.906–4.586
	high/low and low/high vs. low/low	1.412	0.443	0.585–3.404
age (median in years)	>64/≤64	1.290	0.493	0.623–2.675
number of metastases	>1/≤1	1.341	0.418	0.659–2.728
type of metastases	synchronous/metachronous	4.154	**0.004**	1.584–10.893
CD44v6/PD-L1 expression	high/high vs. low/low	2.882	**0.017**	1.213–6.848
	high/low and low/high vs. low/low	0.872	0.723	0.409–1.860

**HR**, Hazard ratio; **PFS**, progression free survival; **CI**, confidence interval; *****, nodules within the metastatic organ; cut-off values defining high- and low-level expression for the individual biomarker are given in Table 3.

## Data Availability

Data corresponding to the analyzed tissues were delivered in anonymized form by the HTCR Foundation.

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
