# Peer review of "High Dual Expression of the Biomarkers CD44v6/α2β1 and CD44v6/PD-L1 Indicate Early Recurrence after Colorectal Hepatic Metastasectomy"

_cancers, 2022, doi:10.3390/cancers14081939_

Round 1

Reviewer 1 Report

The manuscript by Wrana et al., focuses on the determination of biomarkers expressed in metastases of CRC. It is an important issue as metastatic cancer remain largely uncurbable and the choice of treatment based on the biomarkers expression might be a valid option to improve the efficiency of the applied treatments.

Comments:

  1. I believe it would make sense to describe the reasons for the choice of these particular biomarkers in more detail. In the present version it is not clear whether the biomarkers were chosen specifically for the CRC or for advanced tumours in general (“….therapeutic targets in solid cancers”). A rationale for this particular tumour type would be welcome.
  2. Materials and Methods. In the description of the patient cohort there is no mentioning whether any treatments were applied prior to metastasectomy.
  3. Are there any known reasons explaining why biomarkers are usually more expressed in lung rather than liver metastases? Does it also apply to other tumour types?
  4. Moreover, single patients had severely increased expression of a particular marker. Could you link it to more or less sever disease outcome?
  5. Is there any correlation between the expression/intensity of the biomarkers in the surrounding tissue and the disease outcome? According to the M&M section, expression of the markers in the surrounding tissues has also been evaluated, however, no table representing these data have been included.

Minor comments:

I believe the figure/table description should be slightly extended. E.g. I am not sure whether neoadjuvant therapy applies to the patients before or after the surgery etc.

Author Response

Reply to Reviewer 1

Point 1:

“I believe it would make sense to describe the reasons for the choice of these particular biomarkers in more detail. In the present version it is not clear whether the biomarkers were chosen specifically for the CRC or for advanced tumours in general (“….therapeutic targets in solid cancers”). A rationale for this particular tumour type would be welcome.”

Answer 1:

We specified the selection of biomarkers analyzed in this manuscript for colorectal cancer. A rationale for why these biomarkers should be studied in metastatic lesions is provided in the Introduction (lines: 70-83). This includes the addition of new colorectal cancer-specific references and the deletion of general references. 

Point 2:

“Materials and Methods. In the description of the patient cohort there is no mentioning whether any treatments were applied prior to metastasectomy.”

Answer 2:

A detailed description of the first-line treatments can be found in Results, 3.1. patient characteristics (lines: 169-175). Details of neoadjuvant therapy right before metastasectomy are provided in Table 2.

Point 3:

“Are there any known reasons explaining why biomarkers are usually more expressed in lung rather than liver metastases? Does it also apply to other tumour types?”

Answer 3:

Possible mechanisms why certain biomarkers have stronger expression in lung metastases compared with liver metastases are discussed (lines 284-291). We have included some new references such as Garcia-Mulero et al, 2020 (Ref. 59). This work supports the idea that lung metastases are characterized by specific features that are independent of the original cancer type.

Point 4:

“Moreover, single patients had severely increased expression of a particular marker. Could you link it to more or less sever disease outcome?”

Answer 4:

We took a closer look at the outliers of the box plot analysis and compared the extreme values with the survival data (see table below). No correlation was found between severely increased/severely reduced biomarker expression and disease outcome.

Therefore these data were not included in the manuscript.

Point 5:

“Is there any correlation between the expression/intensity of the biomarkers in the surrounding tissue and the disease outcome? According to the M&M section, expression of the markers in the surrounding tissues has also been evaluated, however, no table representing these data have been included.”

Answer 5:

Data obtained in benign liver and lung tissue were summarized in Table S1 and included in the Supplement of the manuscript. No significant correlation was found between the expression of the biomarkers in the benign tissue and disease outcome.

Point Minor comments:

“I believe the figure/table description should be slightly extended. E.g. I am not sure whether neoadjuvant therapy applies to the patients before or after the surgery etc.”

Answer Minor comments:

In the Tables 2,3,4,5 and 6 some variables were concretized in the legend.

Thank you for your comments that significantly improved our manuscript.

Reviewer 2 Report

Friederike, et al report a interesting results that mention the protein expression of CD44v6, dual expression of CD44v6/α2β1, and CD44v6/PD-L1 are predictors of early recurrent in colorectal cancer with liver metastasis, post hepatic metastasectomy patients. It is a valuable idea for treatment plan design in colorectal cancer patients. There are few things need to be clear:

  1. In 2.2. paragraph, please mention who read the IHC slides. How many specialists read the IHC slides. Because IHC is not a quantitative methods for diagnosis and very subjective by different readers. Did you use the machine to read them?
  2. In 2.2. paragraph, line 122-123, authors just put the references describing about ERαv, PR, Her2/new, Muc1, and PD-L1 immunostain cut-off point. How about HGF-R, IGF1-R, EGR-R, CD44v6, α2β1, Hsp90? Is there any scientific/ stastics method to define the cut-off points?
  3. In 2.1. patients cohort. Are all your liver metastasis patient single organ metastasis? Compare with Table 2 and Table 4, does the “number of metastases” mean “nodules in the liver”? or “nodules in different organs”? In your patients cohort, did lung metastasis patients group only have one site, lung, metastasis? Especial for second and later stage relapse patients.
  4. In Table 2, what is neoadjuvant chemotherapy defined in your lung metastasis patients? There are 8 second, and 4 later stage relapse patients in lung metastasis group. Does neoadjuvant chemotherapy mean the chemotherapy prescribed “right before lung metastatic lesion resection”, no matter how many lines therapy? Did those patients have front line chemotherapies? What is the regimen?
  5. In 3.2. line 170-174, is that your results or rational?
  6. In Table 3 and Figure 1, they show cancer cells IHC percentage. How about the surrounding immune cells?
  7. In 3.3 paragraph, please mention the recurrent type (local recurrent, single metastasis, multi-organ metastasis). If patients showed recurrent with metastasis, which organ was occurred?
  8. In Table 5, how to define high or low expression? Do you mean that higher than cut-off point is high and opposite is low? How to find your cut-off point level? As PD-L1 expression, cut-off level are <1, 1-5, 5-10, >10 of CPS in clinical practice, do you think different cut-off level could be useful, instead of high and low, in your case?
  9. In discussion part, line 287, “Addition of the biomarker CD44v6….checkpoint inhibitor therapy”. It is too weak to say that now.
  10. It would be better to know the protein expressions in primary cancer and metastatic tumors, if you can provide. On the other hand, western blot is more quantitive methods to detect protein expression. I would like to see if you can provide it.

Author Response

Reply to Reviewer 2

Point 1

“In 2.2. paragraph, please mention who read the IHC slides. How many specialists read the IHC slides. Because IHC is not a quantitative methods for diagnosis and very subjective by different readers. Did you use the machine to read them?“

Answer 1:

In the revised manuscript, we included a detailed description of the process of biomarker evaluation (2. Materials and Methods, 2.2 Immunohistochemistry and Evaluation of Biomarker Expression, lines 128-136).

Point 2:

“In 2.2. paragraph, line 122-123, authors just put the references describing about ERαv, PR, Her2/new, Muc1, and PD-L1 immunostain cut-off point. How about HGF-R, IGF1-R, EGR-R, CD44v6, α2β1, Hsp90? Is there any scientific/ stastics method to define the cut-off points?”

Answer 2:

In 2. Materials and Methods, 2.2 Immunohistochemistry and Evaluation of Biomarker Expression, lines 140-144 we described the cut-off finding for biomarkers without a reference cut-off point in the literature. In Tables 3 and 5 there is a reference to the cutoff determination in the Materials and Methods section.

Point 3:

“In 2.1. patients cohort. Are all your liver metastasis patient single organ metastasis? Compare with Table 2 and Table 4, does the “number of metastases” mean “nodules in the liver”? or “nodules in different organs”? In your patients cohort, did lung metastasis patients group only have one site, lung, metastasis? Especial for second and later stage relapse patients.“

Answer 3:

In the Results part (3.1. Patient Characteristics) we specified that all liver and lung metastases were diagnosed as single organ metastases. In addition, we concretized that “number of metastases” means “number of nodules in the metastatic organ”(lines 165-167). This definition was further included in the legend of Tables 2,4 and 6.

Point 4:

“In Table 2, what is neoadjuvant chemotherapy defined in your lung metastasis patients? There are 8 second, and 4 later stage relapse patients in lung metastasis group. Does neoadjuvant chemotherapy mean the chemotherapy prescribed “right before lung metastatic lesion resection”, no matter how many lines therapy? Did those patients have front line chemotherapies? What is the regimen?”

Answer 4:

Neoadjuvant chemotherapy means that chemotherapy was administered right before metastasectomy of the tumor sample studied. An explanation was added in the legend of Table 2. A detailed description of first-line treatments can be found in Results, 3.1 Patient characteristics (lines: 169-175).

Point 5:

“In 3.2. line 170-174, is that your results or rational?”

Answer 5:

In the revised manuscript, lines 201-204 (previously: 170-174) represent descriptive results.

Point 6:

“In Table 3 and Figure 1, they show cancer cells IHC percentage. How about the surrounding immune cells?”

Answer 6:

In the present study, in addition to cancer cells, we also examined immune cells for biomarker expression (see Table below). We differentiated between various compartments, namely invasion front, stroma surrounding tumor structures and intratumoral area. However, we feel that these data are beyond the scope of the present manuscript but could be published in an immunological context in the future.

Point 7:

“In 3.3 paragraph, please mention the recurrent type (local recurrent, single metastasis, multi-organ metastasis). If patients showed recurrent with metastasis, which organ was occurred?”

Answer 7:

The follow-up data showed that liver metastases with a high proportion (>30%) of CD44v6+ cancer cells developed multi-organ metastases more frequently than liver metastases with a low (<30%) proportion of CD44v6+ cancer cells. In contrast, no apparent difference was observed between the two groups in the frequency of local recurrence and extrahepatic single metastases. We have included this new finding in the revised manuscript (Section 3.3, lines 234-237).

Point 8:

“In Table 5, how to define high or low expression? Do you mean that higher than cut-off point is high and opposite is low? How to find your cut-off point level? As PD-L1 expression, cut-off level are <1, 1-5, 5-10, >10 of CPS in clinical practice, do you think different cut-off level could be useful, instead of high and low, in your case?”

Answer 8:

In the revised manuscript, we added the explanation that biomarker expression below the calculated cut-off was defined as low expression and biomarker expression above the calculated cut-off was defined as high expression (Section 2.2, lines 142-144 and Table 5, legend). Cut-off point values were determined using the biphasic distribution, which was statistically defined using mean antigen expression in liver or lung metastases (Section 2.2, lines 140-142). In principle, the use of more precise multilevel cut-offs leads to more accurate information. However, in the small sample size cohort studied in this manuscript, two-stage cut-offs might be reasonable.

Point 9:

“In discussion part, line 287, “Addition of the biomarker CD44v6….checkpoint inhibitor therapy”. It is too weak to say that now.”

Answer 9:

In the revised manuscript this discussion point was deleted (lines 334-335).

Point 10:

“It would be better to know the protein expressions in primary cancer and metastatic tumors, if you can provide. On the other hand, western blot is more quantitive methods to detect protein expression. I would like to see if you can provide it.”

Answer 10:

Unfortunately, we cannot implement this interesting proposal. There are several reasons for this: 1) the sample size allocated to us by the local biobank is mostly limited. 2) In one third of the patients, primary tumor samples are not available to us because the primary surgery was performed externally. 3) In this study, snap-frozen metastatic tissue was used, while the available primary tumors are formalin-fixed. However, it is worthwhile to collect these samples for a new project.

Thank you for your comments that significantly improved our manuscript.

Reviewer 3 Report

 Authors demonstrated very important study for colorectal cancer. The fact is that patients who could have hepatic recurrence should be treated with surgery or chemotherapy soon as possible within resectable size. If physicians could detect these patient's recurrence early stage, the outcome could reach satisfied results. Moreover, druggable proteins, CD44v6/alpha2beta1 and PD-L1 were used in this study. It is really meaningful for  interpretation of IHC results and predict progress of medical condition.

Minor revision for authors

1)To prove above mentioned point,

Authors should  explain the procedure of calculating "the percentage of positive tumor cells semi-quantitatively" in Materials and Methods section in P3, line 119 with the name of software or other mechanical items to detect, calculate , and pointing score of pathologically positive cells on IHC slides.

This is important to replicate sophisticated results of authors and apply for clinical stage.

Author Response

Reply to Reviewer 3

Point Minor revision for authors:

“Authors should  explain the procedure of calculating "the percentage of positive tumor cells semi-quantitatively" in Materials and Methods section in P3, line 119 with the name of software or other mechanical items to detect, calculate , and pointing score of pathologically positive cells on IHC slides.

This is important to replicate sophisticated results of authors and apply for clinical stage.”

Answer:

In the revised manuscript, we described in detail the process of detection, calculation, and cut-off determination of biomarker expression (2. Materials and Methods, 2.2 Immunohistochemistry and Evaluation of Biomarker Expression, lines 128-144). In Tables 3 and 5 there is a reference to the cut-off determination in the Materials and Methods section.

Thank you for your comment that significantly improved our manuscript.

Round 2

Reviewer 2 Report

It is pity that you do not have enough samples to do Western blot or other proteomic study. I can understand that. please arrange further study, eg animal study or prospective study in the future.